# Space Emerges from What We Know—Spatial Categorisations Induced by Information Constraints

**DOI:** 10.3390/e22101179

**Published:** 2020-10-19

**Authors:** Nicola Catenacci Volpi, Daniel Polani

**Affiliations:** School of Engineering and Computer Science, University of Hertfordshire, Hatfield AL109AB, UK; d.polani@herts.ac.uk

**Keywords:** spatial cognition, geometric rate-distortion, information theory, information bottleneck, successive refinement

## Abstract

Seeking goals carried out by agents with a level of competency requires an “understanding” of the structure of their world. While abstract formal descriptions of a world structure in terms of geometric axioms can be formulated in principle, it is not likely that this is the representation that is actually employed by biological organisms or that should be used by biologically plausible models. Instead, we operate by the assumption that biological organisms are constrained in their information processing capacities, which in the past has led to a number of insightful hypotheses and models for biologically plausible behaviour generation. Here we use this approach to study various types of spatial categorizations that emerge through such informational constraints imposed on embodied agents. We will see that geometrically-rich spatial representations emerge when agents employ a trade-off between the minimisation of the Shannon information used to describe locations within the environment and the reduction of the location error generated by the resulting approximate spatial description. In addition, agents do not always need to construct these representations from the ground up, but they can obtain them by refining less precise spatial descriptions constructed previously. Importantly, we find that these can be optimal at both steps of refinement, as guaranteed by the successive refinement principle from information theory. Finally, clusters induced by these spatial representations via the information bottleneck method are able to reflect the environment’s topology without relying on an explicit geometric description of the environment’s structure. Our findings suggest that the fundamental geometric notions possessed by natural agents do not need to be part of their a priori knowledge but could emerge as a byproduct of the pressure to process information parsimoniously.

## 1. Introduction

### 1.1. Overview

In the present paper, we set out to take steps towards a programme to characterise, in information-theoretic terms, the question of how an agent can arrive to an understanding of the structure of an agent’s space or environmental “geometry” via only from its interaction with the external world. While such an undertaking is necessarily long term, the present paper begins this by introducing a range of phenomena and issues that we consider relevant to these complex of questions.

Whether the manifold ways in which we suggest geometry manifests itself will indeed be fully or partially expressible in this manner remains to be seen in the future. However, we begin our undertaking by showing that several spatial characterisation phenomena already emerge quite naturally from the information-theoretic characterisation of the problem and this is combined with the typical assumptions in such a context that an agent can only marshal a limited amount of information processing capacity in carrying out its cognitive tasks.

A particular motivation for an information-theoretic treatment is the promise of universality. Such universality is expected to help express and incorporate seemingly disconnected concepts in a common framework. As a step towards this, in the present paper we show how the seemingly purely information-theoretic principle of successive refinement [1] gives rise to an emerging hierarchy of descriptive levels in a manner reminiscent of the seminal hypothesis of the Spatial Semantic Hierarchy [2], but this time it is expressed exclusively in a uniform language, that of information.

The advantage of such an approach is that it would give a better insight into how informational constraints, hypothesised as a core principle of biological information processing [3,4], might drive the cognitive organisation of the organismic perception of space, but also how apparently disparate conceptions may thus emerge simply through informational coarsening.

In summary, the present paper thus sets out towards the following main undertakings: Mapping out the realm of phenomena that are candidates to be considered part of the geometric “understanding” of an agent’s space; making inroads into how to address this question by an informational characterisation of spatial groupings; and finally, showing how, in consequence of this characterisation, one can use the informational principle of *successive refinement* to endow the resulting characterisation with an additional layering, thereby probing how one might be able to understand, from an informational perspective, how different concepts of space coexist on top of or in parallel to each other.

### 1.2. Rationale

We enjoy an excellent mathematical picture of what constitutes our external space. This begins with the insights of Euclidean Geometry, leading ultimately to curved Riemannian spaces which have dominated the treatment of space in the physical world. Not only in physics but also Artificial Intelligence (AI) and robotics are essentially dominated by the view of an external world that is not only objective, but also easily describable in a limited set of minimalistic axioms which impose a significant a priori structure on the concept of space. The latter is amenable to simple and highly regular rules and even (as in curved spaces) when the geometry is not entirely homogeneous, it respects highly constrained principles.

However, that such a regular structure in the concept of space should be pervasive is not altogether obvious. The age and venerability of Euclid’s formulation leads one to forget that it had to be discovered and formalised. Such is the case for curved geometry; the inability to derive the seemingly obsolete Euclidean “parallel postulate” was critical for the discovery that alternatives to Euclidean geometry exist.

Once formalised, this mathematical view of geometry is so dominant that it is difficult to extricate oneself from its perspective. However, this structure of space ceases to be so clear cut in the biological realm. Pigeons are able to travel very long distances without formal knowledge of spherical geometry, relying instead both on global features pervading the environment, such as magnetoreception [5] or local, nongeneralisable features, such as e.g., roads or rivers [6]. The London Tube Map is well known for concentrating on landmarks and topology rather than true distances and angles, and it concentrates on representing which tube lines connect which points on the map—in particular, from the point of view of the Tube Map, “nothing in between stations exists” apart from the connection itself.

Similarly, Polynesian civilisations, which navigated for long distances on the relatively featureless open sea relied on a multitude of features, including swells and currents, sometimes depending on the time of day or year [7]. These features were learnt over decades of a navigator’s lifetime and when encoding information about them in so-called “stick maps”, these features were encoded in a very particular and idiosyncratic way that was not necessarily transferable between navigators, requiring significant individual experience. “Geometry”, from the point of view of such a navigator, becomes a function of the navigator’s own history and “operational map” and is difficult to abstract into an objective entity.

On the other extreme of geometrical reduction, the A1 road from London to Edinburgh in the UK used to be well-known for signage such as “Hatfield and the North” (slightly modified today). It effectively reduces the UK’s geometry to one dimension along the south-north direction, and only mentions the nearest immediate point of interest in that direction once Greater London has been left. The elongated form of the UK, together with the paramount importance of, in particular, the axes out of London lend itself to this simplification. Only once one was branching off the main road, would the finer details of the geometry have to be resolved by a driver.

We have thus seen that the operational treatment of space by agents, be it migrating birds, highway travellers, tube users, or Polynesian seafarers, is quite different from the pure space that mathematicians or, on a more practical level, land surveyors are interested in. Under such considerations, space reveals itself to be a far more operational concept than its mathematical idealisation—the structure of space becomes a function of its use.

When one takes an even closer look at this and considers space strictly through the perspective of the sensorimotor loop as in [8,9,10,11], whatever spatial modalities exist, exist exclusively through what can be sensed and acted upon throughout the closed perception-action loop (see also [12]). The “space” emerging from this interaction is purely defined in terms of what is done vs. what is sensed. Another view on a similar matter, but from a different angle, is based on Integrated Information Theory and superimposes different aspects of “spacehood” in an effectively simultaneous way [13]. Amongst such concepts are location, size, boundary, extendedness, distance, and others. Intuitively,  this captures the fact that, when one speaks about space one has different aspects that can be perceived or manipulated. That these multiple concepts overlap each other is well accepted in mathematics, but rarely explicitly discussed and mostly taken for granted. To illustrate, the set $R^{n}$ is at the same time a set, a topological space, group, vector space, metric space, affine space, (Riemannian) manifold, tangent bundle, and much more. Typically not an observation worth mentioning, the study [13] demonstrates that such overlapping properties may become per se a core component of how an observer or agent forms a representation of its environment as they represent the multi-layered facets characterising an entity, even if that entity is formally completely defined by only few specific postulates. In other words, in the eye of the beholder, space, rather than being an abstract mathematical entity, can be understood as the multiple simultaneously overlapping of properties of the substrate in relation to the agent.

In short, while the structure of space can be in principle compactly encoded in the mathematical formalism using suitable axioms, its various properties are only “unpacked” by agents as they are being used and to different extents. In a proto-mathematical world where Euclid’s axioms were unknown, we postulate that these properties would be utilised as they arise, depending on the relevance to the task at hand. Under this assumption, and the assumption of information parsimony [4] (also see below), we are interested in investigating how such an “unpacking” would take place and which features emerge and in which form, to gain a better understanding about how a picture of a space is formed.

We posit that such a space is necessarily the consequence of a sensorimotor interaction with the world and is thus only defined in terms of such an interaction. In particular, it is not absolute and a priori. While this loses some of the simplicity of the purer mathematical perspectives of spaces described above, it gains the advantage that the concept does not need to talk about physical space, but can instead refer to abstract operations, and more general sensorimotor spaces, whether it is the mastery of a skill, the abstraction of complicated operations (be it puzzle- or game-solving), or other abstracted skills. In other words, by moving towards the sensorimotor perspective (such as [10]), any type of interaction of an agent with its environment defines a type of “space” in a more general sense than just a Euclidean space. The ultimate goal becomes then to understand how different “locations”, i.e., states, in such a space are related to each other in terms of what to do to travel from one point to another and how they are linked together on the meta-level, even when traditional geometry is no longer the appropriate language to discuss this.

Insights from neuroscience [14] indicate that locations are encoded in the brain not just via principles of pure Euclidean geometry, but take into account the constraints on movement through walls. Locations’ encoding may include both a full cognitive graph, or else only dominant routes on overlearning instead of a full representation [15], and the structure induced by geodesic route lengths rather than direct distances [16]. This can be related to models inside the framework of Reinforcement Learning (RL), which consider only actual transitions enacted by an agent, such as in the successor representation [17], and the representation of the space thereunder in suitably chosen bases that suit the particular constraints of available transitions rather than a naive spatial arrangement [18].

We extend this RL-based picture using the aforementioned principle of information parsimony [4,19,20,21,22,23,24]. This principle has been hypothesised to be a trade-off that organisms employ in their strategies, in the sense that decisions are not selected just by optimality, but also in consideration of how much computational (i.e., informational) resources they consume. This is motivated by the observation that information processing is exceedingly expensive and a judicious use of informational bandwidth is essential [3,25]. For modelling, this means that any policy needs to be tempered against the informational cost that it requires to be carried out. We mention that related to the parsimony principle, a Free Energy principle has been postulated as a general principle governing decision-making  [26]. Since our goals are specifically about the representation of near-geometrical task spaces, we will, however, focus on informational constraints imposed on formalisms derived from the RL framework.

Concretely, we treat all geometry essentially as a result of an operational interpretation in the sense of RL. At the same time, we subject this to an information parsimony principle that associates a limited amount of available information processing resources to characterise states thus identified in the world and will study how this affects the way geometry “emerges”.

One aspect we will pay particular attention to will be the question of how a geometrical picture is refined when subsequently more resources are being allocated to the resolution of a space. We believe that this is a crucial step towards an insight on how increasingly sophisticated models are incrementally acquired in a continual learning process and also a key towards modelling the gradual acquisition of the type of multi-layer understanding of space that is proposed e.g., in [13].

Our approach makes use of the information-theoretical notion of successive refinement [1]. Unlike its original use in the service of coding theory, we apply it, to our knowledge for the first time, to the question of how one would increasingly unravel the resolution of a task (in our case, a geometrical) world, as additional informational resources are made available. Our hope is that this will give us an insight on how a hierarchical and increasingly refined picture of a geometrical world can be built up from both the principle of interaction and the principle of information parsimony. Such a systematic approach towards systematically building task space understanding by informational refinement would be helpful to underpin how powerful concepts to model space such as the Spatial Semantic Hierarchy [2] might emerge from first principles.

The world that we study here is limited to symmetric interactions and deterministic transitions only. In other words, it can be modelled as a regular undirected transition graph. The purpose of this is to first fully understand the consequences of trying to approach the geometry of a space with a traditional distance metric from an interaction view in conjunction with an information-reducing perspective.

Our requirement of a symmetrical and deterministic world is adopted here purely to focus on developing the intuition. Note, however, that nothing in formalism limits it to this special case and future work will extend the work to more intricate dynamics that will include probabilistic and nonsymmetric or irreversible dynamics.

The rest of the paper will be structured as follows: In Section 2 we will introduce the proposed model at the basis of our study, together with the mathematical background material; in Section 3 we will present and discuss the results of our numerical investigations; finally, in Section 4 we will draw our conclusions, including a perspectives on future work.

## 2. Methods

In the following we will introduce the Geometric Rate-Distortion (GRD) method (Section 2.1), which is a particular instance of rate-distortion theory [27] where the distortion function quantifies the distance between locations of the space under investigation. GRD constitutes the fundamental building block of our information-theoretic formalism. It will allow us study what kind of spatial structures may emerge from imposing limited information processing to organisms. In addition, we will see that the combination of GRD with the information bottleneck method [28] (Section 2.2) enables the investigation of abstract spatial representations of the agent’s environment. Importantly, utilising only the notion of proximity induced by GRD, the information bottleneck will generate clusters that correspond to interesting features of the environment (Section 3.2). Finally, the successive refinement method [1] will be applied to GRD (Section 2.3) to allow spatial representations to be refined by the agent, instead of necessarily being constructed from the ground up. In Section 3.3, we will show that our formalism permits a notion of space that can be refined, supporting the idea that the ability of natural agents to adapt the complexity of their spatial representations may be understood from an information-theoretic point of view.

### 2.1. Geometric Rate-Distortion

In Information Theory [29,30], the notion of entropy provides a quantification of the amount of information necessary on average to describe a random variable. Given a random variable *X*, with values x∈X, the entropy of *X* is denoted by H(X). When the information contained in *X* is represented by binary codewords, H(X) measures the minimum number of bits necessary to losslessly encode *X*.

In some situations, a perfect description of *X* may not be necessary and an approximate representation that requires less than H(X) bits to describe *X* may be sufficient. Assuming a measure of description fidelity, rate-distortion theory [27,31] addresses the problem of constructing the most compact description $X_{1}^{*}$ of *X*, with values $x_{1}^{*}\in X_{1}^{*}$, given a desired level of fidelity.

In order to quantify the fidelity of $X_{1}^{*}$ in describing *X*, the concept of distortion function is defined as follows:(1)$$d:X\times X_{1}^{*}\to R_{\geq 0}$$
where $d(x,x_{1}^{*})$ measures the cost of using the element $x_{1}^{*}\in X_{1}^{*}$ to represent x∈X. In this paper we will always assume that the event spaces of interest are the same i.e., $X=X_{1}^{*}$, which also implies d(x,x)=0. The minimum number of bits necessary to describe, on average, the random variable *X*, and incurring an average distortion no larger than *D* (i.e.,  a guaranteed minimal fidelity), is called the rate-distortion function R(D). An interpretation of the rate-distortion function is provided in the context of communication systems, where source coding is applied to an independent and identically distributed source *X* that generates a sequence of symbols according to the fixed distribution p(x). In this case the rate R(D) represents the average number of bits per symbol that can be sent in a single transmission through a communication channel, generating on the receiver side an average distortion no larger than *D*. Let $p(x_{1}^{*}|x)$ be the conditional probability of representing the symbol *x* with the codeword $x_{1}^{*}$. Formally, R(D) is defined as the solution of the following constrained optimisation problem:(2)$$R\left(D\right)\;=\;\min_{p\left(x_{1}^{*}|x\right):E\left[d\left(X,X_{1}^{*}\right)\right]\leq D}I\left(X;X_{1}^{*}\right)$$
where $E\left[d\left(X,X_{1}^{*}\right)\right]\;=\;\sum_{x,x_{1}^{*}\in X}p\left(x\right)p\left(x_{1}^{*}|x\right)d\left(x,x_{1}^{*}\right)$ and I(X;Y) denotes the mutual information between the random variables *X* and *Y*, which is defined by $I\left(X;Y\right)≐\sum_{x\in X}\sum_{y\in Y}p\left(x,y\right)\log\frac{p(x,y)}{p(x)p(y)}$. Equation (Equation 2) can be formulated as a Lagrangian optimisation with multiplier $\beta_{RD}$ and solved using the standard Blahut–Arimoto (BA) algorithm [32,33].

The inverse function of R(D) is the distortion-rate function D(R) which, given a rate *R*, is the solution to the following optimisation problem (dual to (Equation 2)):(3)$$D\left(R\right)\;=\;\min_{p\left(x_{1}^{*}|x\right):I\left(X;X_{1}^{*}\right)\geq R}E\left[d\left(X,X_{1}^{*}\right)\right].$$

Typical distortion functions used in information theory are: Hamming distortion, squared error distortion, or absolute error distortion. In tune with the question about appropriate representation of geometry, we here concentrate on distortion functions that measure the geometrical distance between two locations of an environment, in order to investigate the spatial information processed by embodied agents. Given two states $a\;=\;(a_{x},a_{y})\in S$ and $b\;=\;(b_{x},b_{y})\in S$ of a 2D environment (although we focus here on 2D environment only for the sake of clarity, the proposed formalism can be easily extended to multidimensional spaces), the geometric distortion function d(a,b) measures the distance between the two locations. In the context of rate-distortion methods, d(a,b) quantifies the cost of representing the location *a* with the location *b*. We name this geometric interpretation of rate-distortion theory Geometric Rate-Distortion (GRD). The geometric distortion function that will be considered in this paper is the L1 metric (in our study we considered also the L2 metric, which led to comparable results and no particular additional insights). Namely,
(4)$$d_{L1}\left(a,b\right)≐|a_{x}-b_{x}\left|+\right|a_{y}-b_{y}|.$$

In this paper we consider a model of agents that need to represent their own location s∈S with a certain desired level of precision. Given a prior distribution p(s) of locations at which an agent can find itself (we ignore any possible history of movement), and a maximum allowed error *D*, R(D) specifies the minimum amount of information (expressed in bits) needed to represent the agent’s location with an average error from its true position no larger than *D*. In addition, this GRD problem is characterised by the following quantities: The random variable $S_{1}^{*}$, which is the newly compressed description of the possible agent’s locations in the environment; the conditional distribution $p(s_{1}^{*}|s)$, that specifies the probability of representing with the codeword $s_{1}^{*}$ (i.e., an approximate geometric representation) the agent’s position *s*; and the reverse distribution $p(s|s_{1}^{*})$, which indicates the probability of the agent being in position *s* given that the codeword $s_{1}^{*}$ is used as an approximate representation of the agent’s location.

### 2.2. The Information Bottleneck Method

When one applies GRD with a small information rate, the random variable $S_{1}^{*}$ represents a coarsened version of the original agent’s location, effectively a low-resolution version of the original location that requires fewer bits to be specified. As a result, in the coarsened space the environment’s representation is effectively compressed and certain locations may no longer be distinguished. In the present study we use the Information Bottleneck (IB) method [28] as a clustering procedure to group together states that are not distinguishable when an agent represents locations in space approximately.

Consider the two following random variables: The original representation *S* that an agent has about the space of its environment and the compressed geometric representation of this space $S_{1}^{*}$ obtained via GRD. These two random variables are not independent, therefore *S* has some information about $S_{1}^{*}$, a relation that we denote as $S\to S_{1}^{*}$. Given the joint probability distribution $p(S,S_{1}^{*})$ induced by GRD, the aim of the IB method is to construct a "bottleneck" random variable *C* that has access to *S*, but aims to extract information about $S_{1}^{*}$. In other words, *C* extracts information about $S_{1}^{*}$ indirectly through *S*. The IB now aims to extract as much information about $S_{1}^{*}$ as possible, while limiting any superfluous information intake from *S*. We denote this relationship between the random variables *S*, $S_{1}^{*}$, and *C* as $C⇐S\to S_{1}^{*}$.

In the most general case, one can reduce the amount of information sought from $S_{1}^{*}$ as long as the information about *S* can also be reduced. This again becomes a constrained optimisation problem which is expressed as a Lagrangian. It is solved by minimising the following Lagrangian over p(c|s):(5)$$L\left(p\left(c|s\right)\right)\;=\;I\left(C;S\right)-\beta_{IB}I\left(C;S_{1}^{*}\right)$$
where the Lagrangian multiplier $\beta_{IB}$ selects how much information *C* should retain about $S_{1}^{*}$, controlling the trade-off of compression of *S* against preservation of information about $S_{1}^{*}$. The IB method constitutes an information-theoretic generalisation of the notion of sufficient statistics [34]. Note also that the IB can be formulated in terms of a rate-distortion problem with the nonlinear distortion function $d\left(s,c\right)\;=\;D_{KL}\left(p\left(s_{1}^{*}|s\right)\|\;p\left(s_{1}^{*}|c\right)\right)$, where $D_{KL}$ denotes the Kullback–Leibler divergence. According to this view, it can be shown that the IB optimisation problem can be solved numerically with an extension of the BA algorithm introduced in the context of rate-distortion theory [28].

One main application of the IB method is clustering [35], where elements c∈C are used to represent clusters. In the present framework, we will consider the conditional distribution p(c|s) generated by the IB as a “soft partition”: p(c|s) denotes the “probability” that a location *s* will be allocated to cluster *c*. Strictly speaking, this is not a probability, but an allocation weight in our context.

Due to the compression induced by the locations-codewords stochastic mapping of GRD, certain states of the space become undistinguishable. To have an efficient representation of space these states should be grouped together and indexed by dedicated codewords. The first idea would be to utilise the codewords $s_{1}^{*}$ of the compressed space $S_{1}^{*}$. However, using GRD only, the same state *s* may be mapped to more that one codeword $s_{1}^{*}$ with varying levels of uncertainty. Furthermore, as we will see in Section 3.1, the support of the probability distribution $p(S|S_{1}^{*}=s_{1}^{*})$ for a given codeword $s_{1}^{*}$ may have soft boundaries. All these issues make the usage of GRD alone to build a one-to-one relationship between locations and clusters a difficult task. To solve this problem we need a proper clustering method, such as the IB method. In Section 3.2, we will see that the IB $C⇐S\to S_{1}^{*}$ will be capable of constructing such a mapping between locations *s* and codewords *c* where there will be no uncertainty regarding which cluster belongs to which locations and vice versa.

### 2.3. Successive Refinement of Information

We have already mentioned that, using the GRD, we describe the agent’s location at a possibly lower resolution than required to fully resolve its original state. Clearly, the compactness of the representation trades off against its accuracy. Let us now consider an agent that initially operates with a coarse description of the environment. Assume now that circumstances change—either more learning potential becomes available, or a more challenging task emerges—and that the agent now wishes to improve its existing coarse representation of the world. We could, of course, now simply retrain the agent from scratch at the more refined level and recompute a trade-off solution ab initio. However, not only will this be in general quite costly, it is also not plausible to assume that a biological organism would undergo such a complete rearrangement (in fact, discarding and rebuilding) of its learned representation during its lifetime (there may be exceptions for this at certain stages in the life of biological organisms, Refs. [36,37] and we ignore such rearrangements which are triggered by the genetic “program” at particular stages of development of the organism). Rather, we would expect that they would take advantage of the prestructuring and coarse resolution of the already achieved representation. In this spirit and the spirit of life-long adaptation, we propose that a refinement should be based upon this existing representation. Fortunately, the language to express this consistently in the context of information theory is established and known as “successive refinement”.

The successive refinement method [1] has been developed in the context of communication systems to refine the description of a delivered message. Given a coarse description, the refinement essentially consists of an extension of this description to achieve a finer description of the original message. The successive refinement property means that this finer description is not computed from scratch, but, importantly, based on the already existing coarser description. We say that the first description is “successively refinable” when this two-step refined transmission is optimal with respect to the desired distortion level at each stage. We now explain how this applies to our current scenario.

In our context, when agents employ GRD, they do not always need to construct a precise spatial representation from the ground up but can refine an initial description by “appending” an “addendum” of information to the original encoding. In the spirit of continued adaptation, ideally, we aim to retain the same notion of optimality guaranteed by rate-distortion theory for the description obtained when the refinement process is carried out in two successive incremental steps rather than in one go. Formally, given *S*, first rate-distortion is used at rate $R_{1}$ to construct the compressed spatial representation $S_{1}^{*}$ of *S* which leads to an average distortion $D_{1}$. Then, once a coarse location has been chosen, $S_{1}^{*}=s_{1}^{*}$, an addendum of information is constructed at rate $\Delta R≐R_{2}-R_{1}$ (with $R_{2}\geq R_{1}$) such that the two combined representations achieve a total distortion of ${\tilde{D}}_{2}$ (where ${\tilde{D}}_{2}\leq D_{1}$, i.e., the addendum has improved the original distortion). The original space *S* is considered successively refinable if the two following equalities hold:(6)$$R_{1}\;=\;R\left(D_{1}\right)\;\;\;\;and\;\;\;\;R_{2}\;=\;R\left({\tilde{D}}_{2}\right)$$

In other words, the original coarse trade-off, and the secondary finer one based on the first, are both the best that can be achieved for distortions $D_{1}$ and ${\tilde{D}}_{2}$ respectively. Alternatively, let us denote with $D_{2}$ the average distortion incurred when applying GRD to *S* at rate $R_{2}$ (i.e., $R_{2}\;=\;R\left(D_{2}\right)\geq R_{1}$) generates the approximate spatial representation $S_{2}^{*}$ ($D_{1}\geq D_{2}$). Then, Equation (Equation 6) means that the combination of first choosing a coarsely described representation at rate $R_{1}$, and then refining it with an addendum at rate ΔR, leads to the same distortion $D_{2}$ that we would have incurred directly coding the original space *S* at rate $R_{2}$ (i.e., ${\tilde{D}}_{2}\;=\;D_{2}$). In this regard, an equivalent definition can be formulated in terms of distortion-rate functions. Let us denote with ${\tilde{S}}_{2}^{*}$ the spatial representation obtained refining $S_{1}^{*}$. Then, *S* is successively refinable if the following inequalities hold:(7)$$E\left[d\left(S,S_{1}^{*}\right)\right]\leq D\left(R_{1}\right)\;\;\;\;and\;\;\;\;E\left[d\left(S,{\hat{S}}_{2}^{*}\right)\right]\leq D\left(R_{2}\right).$$

Equitz and Cover [1] provide a Markovian elegant characterisation that can be used to determine whether successive refinement is achievable. According to their criterion successive refinement with distortions $D_{1}$ and $D_{2}$ is achievable if and only if, for all $s,s_{1}^{*}$ and $s_{2}^{*}\in S$,
(8)$$p\left(s_{1}^{*},s_{2}^{*}|s\right)\;=\;p\left(s_{2}^{*}|s\right)p\left(s_{1}^{*}|s_{2}^{*}\right)$$
where Equation (Equation 8) implies that the random variables $S,S_{1}^{*}$ and $S_{2}^{*}$ form the Markov chain $S\to S_{2}^{*}\to S_{1}^{*}$. Equation (Equation 8) can be expressed in information-theoretic terms using the following formulation: Random variables $S,S_{1}^{*}$ and $S_{2}^{*}$ form the Markov chain $S\to S_{2}^{*}\to S_{1}^{*}$ if and only if:(9)$$I(S;S_{1}^{*}|S_{2}^{*})\;=\;0.$$

In Section 3.3 we will use Equation (Equation 9) to determine whether for pairs of GRD’s rates $\left(R_{1},R_{2}\right)$ successive refinement is achievable. Furthermore, we will present an experiment where the successive refinement method will be applied to a simple environment using Algorithm 1. This procedure can be utilised to compute ${\tilde{D}}_{2}$ and ${\tilde{S}}_{2}^{*}$ given p(S), $R_{1}$, and ΔR. The computation of ${\tilde{S}}_{2}^{*}$ proceeds as follows (see Algorithm 1 for more details): First, $S_{1}^{*}$ is obtained compressing the original space *S* at rate $R_{1}$ via GRD (line 1); then, $S_{1}^{*}$ is refined for each codeword $s_{1}^{*}$ of $S_{1}^{*}$ separately using GRD at rate ΔR (line 4); finally, the resulting distribution $p(S_{2}^{*}|S,S_{1}^{*})$ is averaged over p(S) and $p(S_{1}^{*})$ to obtain $p(S_{2}^{*})$ (lines 6 and 7).
**Algorithm 1**$\left[p\left({\tilde{S}}_{2}^{*}\right),{\tilde{D}}_{2}\right]$ = SuccessiveRefinement($p\left(S\right),R_{1},\Delta R$)1:$\left[p\left(S_{1}^{*}\right),p\left(S_{1}^{*}|S\right)\right]\;=\;GRD\left(p\left(S\right),R_{1}\right)$2:$p\left(S|S_{1}^{*}\right)\;=\;\frac{p\left(S_{1}^{*}|S\right)p\left(S\right)}{p(S_{1}^{*})}$3:**for** $s_{1}^{*}\in S$**do**4:    $\left[p\left({\tilde{S}}_{2}^{*}|S_{1}^{*}=s_{1}^{*}\right),p\left({\tilde{S}}_{2}^{*}|S,S_{1}^{*}=s_{1}^{*}\right)\right]\;=\;GRD\left(p(S|S_{1}^{*}=s_{1}^{*}),\Delta R\right)$5:**end for**6:$p\left({\tilde{S}}_{2}^{*}|S\right)\;=\;\sum_{s_{1}^{*}\in S}p\left({\tilde{S}}_{2}^{*}|S,S_{1}^{*}\;=\;s_{1}^{*}\right)p\left(s_{1}^{*}\right)$7:$p\left({\tilde{S}}_{2}^{*}\right)\;=\;\sum_{s\in S}p\left({\tilde{S}}_{2}^{*}|S\;=\;s\right)p\left(s\right)$8:${\tilde{D}}_{2}\;=\;E\left[d\left(S,{\tilde{S}}_{2}^{*}\right)\right]\;=\;\sum_{s,s_{2}^{*}\in S}p\left(s\right)p\left(s_{2}^{*}|s\right)d\left(s,s_{2}^{*}\right)$9:**return** $p\left({\tilde{S}}_{2}^{*}\right),{\tilde{D}}_{2}$

The application of the successive refinement method to GRD will allow us to investigate whether there are spatial representations with resolutions that are more favourable than others (Section 3.3). Indeed, agents that compress their space at the information rates where successive refinement is achievable have the advantage of not building all their descriptions from the ground up. Instead, using successive refinement, they can switch from a coarse representation to a more sophisticated one utilising minimum informational effort.

## 3. Results and Discussion

### 3.1. Spatial Compression via Geometric Rate-Distortion

In this section we investigate the spatial representations induced by GRD for two 10 × 10 grid worlds characterised by the following topologies: A single room environment and a multi-room environment composed by four 5 × 5 rooms connected by passages situated at the centre of the dividing walls. To illustrate the spatial approximation carried out by GRD, assuming that the exact location of the agent *s* is uniformly distributed, we report the GRD output distribution $p(S_{1}^{*})$, that characterises the adopted codebook indicating the probability of using symbols $s_{1}^{*}$ to describe agent’s locations. In addition, we report the source coding distribution $p(S_{1}^{*}|s)$, which indicates the probability of representing the agent’s position *s* with the codeword $s_{1}^{*}$. Finally, the conditional probability distribution $p(S|s_{1}^{*})$ will allow us to estimate the uncertainty about the agent’s location when its position is represented with the symbol $s_{1}^{*}$.

In Figure 1 we focus on a single-room environment, reporting the aforementioned probability distributions for the three $\beta_{RD}$ values 0.1, 0.65, and 15. In the figures, the cells within the grid world indicate agent locations within the environment *S* or codewords locations within the approximate spatial representation $S_{1}^{*}$. Cells are coloured according to the probability of the reported distributions, with values indicated by the colour bars. For $\beta_{RD}\;=\;0.1$, every location of the environment is mapped according to a uniform distribution to one of the four codewords located at the centre of the environment. This can be seen in Figure 1a, which depicts the prior probability of picking a codeword $p(S_{1}^{*})$ and shows the four yellow states standing out each one with probability 0.25. In addition, a similar probability distribution can be observed in Figure 1d, where $p(S_{1}^{*}|s)$ is represented when the agent position is in the top left location of the grid world ($s\;=\;\left(s_{x},s_{y}\right)\;=\;\left(1,10\right)$, marked with a magenta circle). With such a small $\beta_{RD}$, the restriction imposed by the the geometric constraint is almost negligible, leading to an average distortion of D=4.965, which is basically the average distance of the codeword (distribution) to any possible candidate location in the grid. The information shared by the original and compressed spatial representations, measured by the rate *R*, is almost zero (R=0.001 bits). In other words, having one of these four codewords selected tells us little about the real position of the agent and does not contribute to effectively decreasing the distance cost as opposed to not knowing it. This is reflected by $p(S|S_{1}^{*}=s)$ (Figure 1g), which shows that to use the codeword $s_{1}^{*}\;=\;\left(5,6\right)$ (represented by a red circle) induces a large uncertainty about the real position of the agent.

When the geometric constraint becomes more stringent (D=3.287 with $\beta_{RD}\;=\;0.65)$) codewords are more representative of the original agent’s location (R=0.789 bits). In fact, in Figure 1b $p(S_{1}^{*})$ shows that codewords with positive probability are more spread out and on average are placed nearer to all individual states of the environment instead of being in the middle of it. Furthermore, codewords are not selected with the same probability distribution for all agent locations, as for $\beta_{RD}\;=\;0.1$, but codewords that distribute possible agent locations amongst themselves are favoured. This is shown in Figure 1e, where $p(S_{1}^{*}|S=\left(1,10\right))$ shows a larger probability for codewords that are closer to the given agent’s position. We also observe that the probability mass of $p(S|S_{1}^{*}=\left(3,8\right))$ (Figure 1h) is concentrated around the given codeword, showing how a larger rate and a smaller distortion reduces the uncertainty about localisation and implies a more faithful approximation of the original space. In this regard, when $S_{1}^{*}$ is permitted to use nearly all the information necessary to fully describe all the positions of the grid world (R≃H(S)=6.642 for $\beta_{RD}\;=\;15$), a near perfect match between codewords and environment locations is obtained (see Figure 1f,i), which leads to a negligible average distortion of D=0.001. In this extreme case, the distribution of codewords $p(S_{1}^{*})$ is uniformly distributed, with every symbol representing one of the original locations (and obviously distributed with a probability value of $\frac{1}{100}$), as shown by Figure 1c.

Above, we have investigated the relationship between rate and average geometric distortion for three values of $\beta_{RD}$ only. In Figure 2, we report how the rate generally depends on the distortion for the aforementioned grid world. As for classical rate-distortion problems, the trade-off curve shows that the rate is a monotonically decreasing function of the distortion. When the average distortion is 0 the permitted rate reaches H(S), which represents the full uncertainty over the agent’s location. Furthermore, when the distortion is maximum the rate is 0, meaning that no information is available to represent the agent’s position.

The relation between information and distortion in general is well-studied, but, in our case, we are interested in the geometrical consequences of such a reduction of information. The encoding shows that, at intermediate levels at least, the codewords capture aspects about the spatial structure of the agent’s world, splitting it into different areas for which different codewords are representative. The coherence of these areas is induced purely through the distance-based distortion. Information theory on its own has no way of incorporating this spatial information into its structure, as in the pure information-theoretic view symbols are transmitted purely based on their probabilistic profile, not based on their similarity.

In Figure 3, an environment with a more complex topology is considered. The 10 × 10 grid world is divided by walls (depicted by burgundy lines) into four rooms of size 5 × 5. In this regard, the distortion function is modified taking into account the fact that agents cannot move through walls. In Figure 3a, $p(S_{1}^{*})$ is reported for $\beta_{RD}\;=\;0.1$ (R=0.028 bits). As for the single room scenario, we have a uniform distribution of codewords. In this environment eight codewords are conveniently placed across the four doors. The average distortion (D=5.668) takes into account the presence of walls, which increases the average distance between locations of the room. With $\beta_{RD}\;=\;0.65$ a larger rate (R=0.825) and a smaller distortion (D=3.317) are achieved. The distribution $p(S_{1}^{*})$, depicted in Figure 3b, shows that codewords are concentrated at the centre of the rooms. These codewords are the closest to all the locations within the rooms and have the largest probability to be chosen. This is characteristically different from the near-zero rate case. When the agent is in location s=(5,6), in $p(S_{1}^{*}|S=s)$ the nearest codewords to its location are favoured to represent its position (see Figure 3e). Finally, in Figure 3h, we see that when the codeword $s_{1}^{*}\;=\;\left(3,8\right)$ is used, the distribution of the inferred agent’s location $p(S|s_{1}^{*})$ is concentrated around the codeword location. The support of this distribution, when compared with Figure 1h, tends to be more allocated around the room’s edge, reflecting the modified geometry of the environment.

The codewords’ distributions of Figure 1 and Figure 3 show that the limitation on agents’ information processing imposed by GRD enables rich geometrical representations of the environment to emerge naturally. In fact, the simple notion of distance embedded in the distortion function provides to codewords geometric features, which give to those symbols a semantics that usually classical information-theoretic representations lack.

### 3.2. Spatial Clustering via Information Bottleneck

The previous GRD models, while limiting the encoding information, still permitted a rich set of coding symbols. In fact the available symbol set was as rich as the original set of locations, even if the information was not necessarily fully faithfully represented. However, in general that will not be the case. If we indeed have to compress, we need to consider what happens in a drastically impoverished representation, which we model inside the framework of information theory via the Information Bottleneck formalism.

When agents possess limited information to represent the space around them, certain locations become undistinguishable. We have seen this in the context of GRD: When the available rate is low and a specific codeword is given, more than one location is inferred with a nearly uniform probability distribution (for instance, see Figure 3h). In this section we investigate how similar informational constraints allow agents to construct spatial representations that group together these undistinguishable states. We will adopt the IB method to partition the space of the environment using the proximity estimates induced by GRD. We will investigate the role of the GRD’s rate in shaping the clustering obtained by the IB $C⇐S\to S_{1}^{*}$.

Crucially, we will study whether this clustering induces partitions that are similar with what we would expect from an efficient and natural way to represent space, taking into account interesting features of the environment such as floors, lifts, and rooms of a grid world-like building.

In Figure 4, we report the results obtained for a 10 × 10 grid world using four possible clusters (|C|=4) and three different values of the Lagrangian multipliers $\beta_{RD}$ and $\beta_{IB}$, which are selected independently for the rate-distortion and bottleneck (in the case of the bottleneck, they indicate how much emphasis one lays on the compression of the original—in our case, coarsened—information into the IB clusters). Each cell of the grid is coloured according to the distribution p(C|S), describing the probability that a location *s* belongs to a cluster *c*. For $\beta_{RD}\;=\;0.1$ and $\beta_{IB}\;=\;1$ (Figure 4a), we observe that all states are included in each cluster with a uniform probability distribution. This is caused by the reduced role that the environment’s topology has on the clustering process when the magnitude of the two multipliers is low. In fact, a small $\beta_{RD}$ implies that a less stringent geometrical constraint is imposed on $S_{1}^{*}$. In addition, a small $\beta_{IB}$ implies that the information about $S_{1}^{*}$ squeezed out by the IB from *S* into *C* is small. Note that the IB obtains the geometrical relationship between the states of the environment from the conditional probability distribution $p(S_{1}^{*}|S)$ and without that information the locations of these states would be irrelevant for the IB. Increasing the Lagrangian multipliers to $\beta_{RD}\;=\;0.6$ and $\beta_{IB}\;=\;3$ (Figure 4b) we see a bifurcation into four soft clusters, which coarsely partition the space in four areas of more or less equal size. In Figure 4c, $\beta_{IB}$ is increased to 5, resulting in a transition from a soft clustering to a harder one. Finally, with $\beta_{IB}\;=\;25$ (Figure 4d), a fully hard partition with clusters of equal size is obtained. Note that in all presented cases, locations that belong to the same cluster are adjacent to each other and without the geometrical information injected in *C* from $S^{*}$ (e.g., when $S^{*}$ is a perfect representation of *S* for $\beta_{RD}\to \infty$), the IB would position the cluster’s elements in random locations of the environment without employing any notion of proximity.

In Figure 5, a more complex topology is investigated where a 32 × 32 grid world has been divided in four rooms of different size, connected by one cell-sized open passages. We intentionally chose an asymmetric set up here to highlight features that would be unremarkable in a symmetric set up, such as in the previous Figure 4. Given that the average distance between locations that belong to different rooms is longer than the average distance between those that are located in the same room, the IB tends to cluster together states within rooms. This result is reminiscent of [38], but with the difference that here we utilise directly the IB together with the GRD coarsening of locations to see this effect (in [38], the room decomposition was derived from a more intricate goal-relevance representation). This can be seen in experiment (a), where with $\beta_{RD}\;=\;0.6$ and $\beta_{IB}\;=\;3$, each cluster is assigned to one room, as we would expect from a plausible and efficient spatial representation of this environment. Note the uncertainty in the locations around the doors, which are categorised as belonging to more than one cluster, namely to the two clusters corresponding to the rooms that the door connects (similarly to what has been reported in experiments (b) and (c) of Figure 4).

In general, such a correspondence between the environment’s topology and the informationally extracted clusters is not always guaranteed but depends on a set of concomitant factors. First, when the IB method is permitted to use less information to compress *S* and has more leeway to minimise I(S;C), it prefers to partition the space in clusters of equal size, which may not correspond to the given environmental topology (this highlights the aforementioned probabilistic rather than geometrical nature of the informational perspective). This happens when the constraint about preserving $S_{1}^{*}$ is less stringent (corresponding to a smaller $\beta_{IB}$), and it also implies that *C* extracts less geometric information from $S_{1}^{*}$.

Another factor to take into account is the support of the probability distribution $p(S|S_{1}^{*})$. Importantly, if $R_{RD}$ is too large, the support of this distribution becomes more localised in the area around $s_{1}^{*}$ and consequently long-distance geometric relations may matter less or may be entirely lost. In the case of the multi-room scenario, this phenomenon could make central states unrelated to the room’s edges. In short, if there is too much informational precision, the geometric representation actually loses—in this framework—insight into the spatial coherence of the environment. In other words, imprecision to some extent is in fact central to establish the spatial coherence and structure of the world. This phenomenon is related to the regularisation properties of the description of the world, but here it is described in purely informational terms (of course, with the distortion term representing the geometric aspect of the problem). In experiment (b) of Figure 5, we present an example of what happens when we coarsen both the geometric information (increasing $\beta_{RD}$) as well as reducing the information that the bottleneck is permitted to extract (decreasing $\beta_{IB}$). In the figure we see that a cluster composed of two rooms crosses the rooms’ boundaries, not fully reflecting the rooms’ topology, and another room is represented by two clusters instead of one.

Figure 6 and Figure 7 show the results of the proposed clustering procedure for environments with a more complex topology. The Lagrangian multipliers $\beta_{RD}$ and $\beta_{IB}$ have been selected with the aim of obtaining a clustering that reflects the environment’s topology, also taking into account the considerations mentioned above in this regard. In Figure 6 we report the results obtained from the clustering of a 45 × 45 grid world composed of 16 rooms for the cases |C|=2,4,8 and |C|=16. These are explicitly selected, but a selection method such as in [38] could also be used to identify preferred codebook sizes. In the figure, a different colour is assigned to each cluster and each state *s* is coloured with a certain colour if this belongs to the corresponding cluster *c* with probability close to 1 (i.e., p(c|s)≃1). We observe that the obtained partitions respect the multi-room topology in all the reported cases. The procedure groups together adjacent rooms in the same partitions when the number of available clusters is smaller than the number of rooms and it assigns one cluster per room when these two sets have the same cardinality.

In Figure 7, a different topology is considered describing a schematic representation of a multi-floor building. The building is composed of ten 10 × 10 grid world-like floors, each one connected to the others by a lift. In every floor, the entrance of the lift is located in the middle of the floor’s left edge (i.e., $s_{lift}\;=\;\left(1,5\right)$), as highlighted in the figure by blue-framed squares. Moving from one floor to another does not happen instantaneously, but it takes a certain amount of time which, to be incorporated into the environment’s geometry, is converted into three units of distance for each traversed floor. Specifically, the lift is represented by a single state connected to the lift’s entrances of all the floors and, once in the lift, to move from the *i*th-floor to the *j*th-floor costs a distance of 3|i−j|. From the resulting clustering, with $\beta_{RD}\;=\;0.6$ and $\beta_{IB}\;=\;35$, not only we observe that we get a separate cluster for each floor, but the lift, being a special location that connects several areas of the environment, is allocated to a dedicated separate cluster which comprises also the areas that surround the lift’s entrances of every floor.

The results reported in Figure 6 and Figure 7 show that the combination of GRD and IB partitions the space in a way that is similar to what we would expect from a compact and plausible spatial representation for the considered grid worlds, assigning clusters to features of the environment that are relevant from a geometrical perspective. This suggests that the ability of natural agents to abstract the environment’s space at different levels of resolution may be a result of their limited resources in terms of information processing.

### 3.3. Successively Refinable Spatial Representations

Until now, much of what we have considered can be viewed as traditional geodesic distance-based clustering, merely cast into an information-theoretic language. While it is obviously useful to be able to express things in an information-theoretic language, the real advantage of this step lies in its universality and a very well-defined concept of separation of structural assumptions (such as, in our case, the distortion) and generic information-processing assumptions. Not only that, expressing the problem in the information-theoretic framework permits us to plug different concepts seamlessly together, as long as they are all ultimately expressed in the information-theoretic language. So we now proceed to take specific advantage of the information-theoretic formulation to gain insight of how and when different levels of description refinement can be related to each other, expressed in the language of the successive refinement method described earlier.

We remind the reader that we consider agents who may need to adapt the precision of their spatial representations because some information that was considered unnecessary in the original learning phase suddenly becomes necessary. To this end, it would be desirable that they would represent their environment adopting descriptions that are capable of being refined without having to relearn the already achieved representation. This would allow agents to switch from one approximate representation to a more precise one, successively refining the first one to obtain the second instead of constructing the latter from the ground up. In this section we will study the feasibility of generating successively refinable spatial representations in the context of GRD formalism. In addition, we will present a simple example of such a geometric successive refinement.

A striking feature of GRD is that spatial representations with certain levels of resolution can be successively refined optimally (as per the Equitz–Cover criterion [1]), whereas other levels of resolution can still be successively refined to an extent that is close to such notion of optimality. As discussed in Section 2.3, successive refinement is said to be achievable when the average distortion ${\tilde{D}}_{2}$, incurred in refining a spatial representation described with rate $R_{1}$ using an addendum of information described with rate $\Delta R≐R_{2}-R_{1}$, is the same distortion $D_{2}$, incurred when the original representation is described directly with rate $R_{2}$. Where [1] provides a characterisation of successive refinement in terms of Markovianity between codeword distributions described at different rates, the performance of the successive refinement method when there is no full Markovianity has, as far as we are aware, not been studied yet. Since the relationship between *S*, $S_{1}^{*}$ and $S_{2}^{*}$ is not in general fully Markovian, in this section we investigate numerically the feasibility of using a relaxed version of the successive refinement method for GRD when $I(S;S_{1}^{*}|S_{2}^{*})$ is close to 0. By relaxed successive refinement here we mean that instead of strictly requiring ${\tilde{D}}_{2}\;=\;D_{2}$ we also permit ${\tilde{D}}_{2}≳D_{2}$, i.e., suboptimal refinements.

In Figure 8, we compare the Markovianity of $S\to S_{2}^{*}\to S_{1}^{*}$ with the error $\Delta D≐{\tilde{D}}_{2}-D_{2}$ (describing the suboptimality of the relaxed refinement) for a 10 × 10 grid world. To this end, we embed both $I(S;S_{1}^{*}|S_{2}^{*})$ and ΔD in the $\left(R_{1},R_{2}\right)$ plane. In Figure 8a we observe that full Markovianity (i.e., $I(S;S_{1}^{*}|S_{2}^{*})\;=\;0$) is rarely obtained for GRD in this environment, namely when $R_{1}$ is very small and $R_{2}$ is large (i.e., when a large refinement is applied to a very coarse representation). If we consider a soft version of Markovianity, we see that the area where $I(S;S_{1}^{*}|S_{2}^{*})\ll 1$ can be made wider, depending on tolerance, and extends toward larger values of $R_{1}$ and smaller values of $R_{2}$. The geometrical descriptions becomes less Markovian when $R_{1}$ and $R_{2}$ have a similar magnitude (i.e., as the refinement is done with small ΔR). Finally, when both rates approach $\frac{1}{2}H\left(S\right)$, around the centre of the $\left(R_{1},R_{2}\right)$ plane’s diagonal, $I(S;S_{1}^{*}|S_{2}^{*})$ reaches its peak. In short, splitting the contributions of the refinement levels violates the Markovianity most strongly.

Looking at Figure 8b we see that ΔD has a similar landscape, apart from the fact that the area with maximum error is not only centred around $\left(\frac{1}{2}H\left(S\right),\frac{1}{2}H\left(S\right)\right)$ but it starts from there and, interestingly, extends up to H(S). In other words, it is not the area with a maximal violation of Markovianity that incurs the most drastic cost inefficiency. We also observe that for pairs of rates where the geometric representations are fully Markovian, we also have ΔD=0, as prescribed by Equitz and Cover. Interestingly, according to our numerical experiment, ΔD is close to zero (i.e., ΔD<0.001) for an area larger than the one where full Markovianity is met, showing that a weaker version of Markovianity may be sufficient to achieve relaxed successive refinement. Finally, we see that as we move far from the $\left(R_{1},R_{2}\right)$ plane’s diagonal, ΔD steeply descends towards values close to zero where relaxed successive refinement becomes achievable. We conclude that, while in this environment strict successive refinement is achievable only for a small area of the $\left(R_{1},R_{2}\right)$ plane, a relaxed version of successive refinement is achievable for a relatively large area of the plane. In other words, if slight compromises are permitted, a reasonably good successive refinement cascade, at least with one intermediate step, is achievable.

In Figure 9, we present a concrete example of relaxed successive refinement. For $R_{1}=0.402$ bits and $R_{2}=4.643$ bits we have $I(S;S_{1}^{*}|S_{2}^{*})=0.022$ bits, hence, according to our hypothesis, relaxed successive refinement should be achievable. First, GRD is applied to *S* at rate $R_{1}$ to construct a description $S_{1}^{*}$ with distortion $D_{1}=3.959$. Figure 9 shows that when locations are represented with $R_{1}$ bits and the codeword $s_{1}^{*}=\left(5,6\right)$ is used (indicated by a red circle), the probability of the inferred position of the agent $p\left(S|S_{1}^{*}\right)=\left(5,6\right)$ is quite uncertain. This coarse spatial representation is then improved with successive refinement using an addendum of information of ΔR=4.241 bits. The resulting spatial representation ${\tilde{X}}_{2}^{*}$ incurs in a total average distortion of ${\tilde{D}}_{2}=0.470$. The precision gained via successive refinement can be observed looking at the probability distribution $p(S|{\tilde{S}}_{2}^{*}=\left(5,6\right))$, which has a significantly smaller entropy than $p(S|S_{1}^{*}=\left(5,6\right))$. Finally, directly applying rate-distortion to *S* at rate $R_{2}$, a description $S_{2}^{*}$ with distortion $D_{2}=0.451$ is obtained, which is very close to the distortion obtained with successive refinement ${\tilde{D}}_{2}$ (ΔD=0.019). Hence, for this particular $\left(R_{1},R_{2}\right)$ pair relaxed successive refinement is achievable. This result is also confirmed by looking at the distribution $p(S|S_{2}^{*}=\left(5,6\right))$ reported in Figure 9, which is almost identical to $p(S|{\tilde{S}}_{2}^{*}=\left(5,6\right))$.

In this section we have shown that successive refinement is achievable, fully or in its relaxed interpretation, for a simple 10 × 10 grid world. This open the interesting question whether the ability of natural agents to adapt efficiently the resolutions of their spatial representations according to the current circumstances can be explained in information-theoretic terms. We will further discuss the consequences of this promising result in the next conluding section.

## 4. Conclusions

In the quest to understand the emergence of notions of space, and inspired by sensorimotor accounts of contingency [8] as well as the multilayered nature of spatial properties [13], we employed an operational model of a task space, based on the Reinforcement Learning framework [18], to specify a geometry under the additional perspective of limited information. We asked how a constrained information capacity would give rise to characteristic spatial features, for instance the emergence of rooms. While straightforward clustering and min-cut algorithms are seemingly natural choices for splitting up graphs, they rely on an intuition about how graphs should be decomposed. Here we reduced assumptions based on a particular graph representation and instead adopted an informational approach to do so which takes into account the information required to specify a region in the space of operation. We proceeded with an approach that is related to [38], but, instead of considering goals retained in limited working memory, we obtained the representation for states to be distinguished using a compression via a rate-distortion-type scheme. While we applied it to a symmetric effective graph distance model, nothing in this scheme limited us to symmetric distances or to distance costs. Everything transferred directly to general RL scenarios, permitting more intricate spatial structures to be extracted. This allowed the intrinsic emergence of “rooms” (without the concept having to be explicitly defined ahead of time), as well as functionally coherent parts of task space (elevator zone).

In the geometric rate-distortion formalism presented here actions were defined implicitly as those operations that allowed agents to directly move from one state to another. An interesting extension of GRD would be to introduce actions explicitly, as for standard RL. This would allow, for instance, to investigate the relationship between the approximate spatial representations obtained with GRD and different agent embodiments (e.g., mapping actions to spatial directions). In addition to actions, the introduction of stochastic transitions would enable one to combine GRD with the full RL framework beyond the successor representation of [17].

As an important consequence of the rate-distortion approach employed here, we were able to investigate the emerging structures under the perspective of information-theoretic successive refinement. This model is, to our knowledge for the first time, applied here to decision-making scenarios. We showed that there were regions of the rate-distortion regime, i.e., the trade-off domain between resolution and required information load, where a coarser representation could seamlessly be refined by adding information, without having to reorganise it. The zone of perfect refineability (without losses) was relatively tight according to Equitz and Cover’s Markovian criterion, however, for a relaxed version of Equitz and Cover’s criterion, one gained a much larger area where an approximate refineability was still possible. It means that the original coarser level could be further refined without rearranging the already established clustering.

Compared to a “re-clustering”, this meant that the framework of successive refinement may offer a route towards a more organic continued acquisition of features that does not have to be relearnt from scratch as more information comes in. The Equitz and Cover criterion permits one also to identify when this is (either precisely or approximately) possible. We believe that this opens up a path for a more systematic approach to not only continual learning, but also to the natural, rather than imposed, emergence of hierarchies describing the system under consideration. Possibly, in the future, it may help to also understand better how the concept of a given space and its potentially multilayered nature could emerge via an agent’s interaction with it and “tuning” through the thus arising informational levels of description.

Although in this paper we presented the successive refinement method with two levels of refinements, the introduced formalism could be naturally extended to implement additional chained refinements. In future work, we would like to exploit the tree structure underlying the successive refinement method to obtain a cascade of spatial coarsenings linked by Markovian or near-Markovian relations. Then, when sequences of refinements are introduced, the presented formulation could be extended towards a hierarchy of refinements. This formalism could show that the hierarchical representations employed by organisms may emerge from information-theoretic principles and that, for instance, the concept of topology may be a consequence of having spatial representations with different levels of resolution.

While the combination of GRD with the successive refinement method induced hierarchies embedded in an informational space, we have seen that, when GRD was combined with the information bottleneck method, abstractions (i.e., clusters) were obtained within the space of the environment. In principle, we would expect it to be possible to continue this abstraction process towards higher levels of representation where new abstract states would be given by clusters. In future we would like to combine the hierarchies embedded in the informational space of the successive refinement method with the ones embedded in the environmental space of the information bottleneck method and vary the two corresponding levels of abstractions in parallel. This might offer a route towards a rich account of the different processes involved in organisms’ spatial cognition when these adopt hierarchical representations. Crucially, we believe that the application of the successive refinement method to the full RL framework could introduce a new perspective on models of hierarchical planning and learning, where the informational cost of switching from one abstract representation to a more refined one (and vice versa) could be taken into account and investigated.

## Figures and Tables

**Figure 1 entropy-22-01179-f001:**
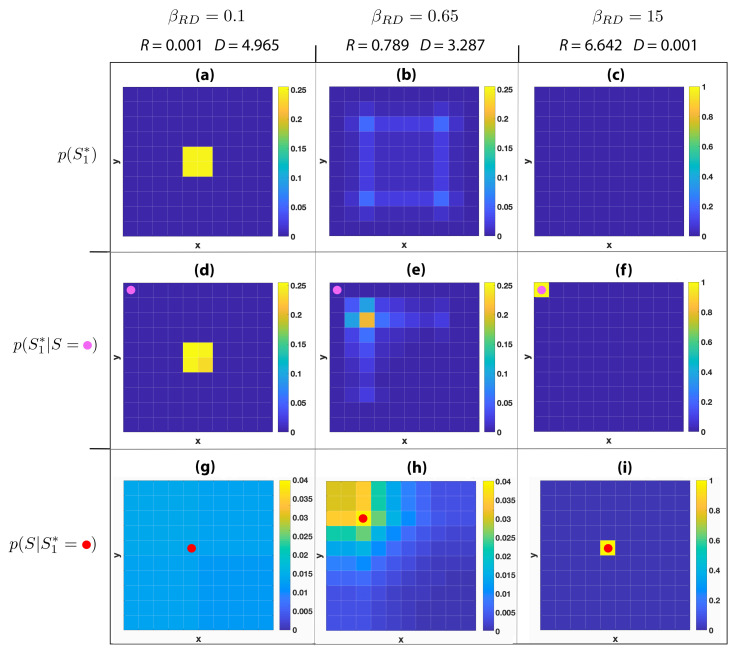
Different probability distributions resulting from the application of geometric rate-distortion to a 10 × 10 grid world. Cells are coloured according to the probability of the reported distributions, with values indicated in the colour bars. Each column represents a different level of trade-off for $\beta_{RD}\;=\;0.1$, 0.65, and 15, where small $\beta_{RD}$ implies a coarse spatial representation and large $\beta_{RD}$ implies a more accurate one. For each trade-off level, the corresponding rate *R* and distortion *D* are reported. In the first row cells are coloured according to the prior probability of choosing a codeword $p(S_{1}^{*})$. The second row depicts values of $p(S_{1}^{*}|S\;=\;s)$, which is the probability of using a codeword given that the agent is in location *s*, where the location s=(1,10) is represented in the grid by a magenta circle. The third row reports $p(S|S_{1}^{*}\;=\;s_{1}^{*})$, that is the probability of the agent being in a certain state given that the codeword $s_{1}^{*}$ has been selected, where the position of $s_{1}^{*}$ is indicated by a red circle. (**a**) $p(S_{1}^{*})$ for $\beta_{RD}\;=\;0.1,R=0.001$ and D=4.965; (**b**) $p(S_{1}^{*})$ for $\beta_{RD}\;=\;0.65,R=0.789$ and D=3.287; (**c**) $p(S_{1}^{*})$ for $\beta_{RD}\;=\;15,R=6.642$ and D=0.001; (**d**) $p(S_{1}^{*}|S\;=\;\left(1,10\right))$ for $\beta_{RD}\;=\;0.1,R=0.001$ and D=4.965; (**e**) $p(S_{1}^{*}|S\;=\;\left(1,10\right))$ for $\beta_{RD}\;=\;0.65,R=0.789$ and D=3.287; (**f**) $p(S_{1}^{*}|S\;=\;\left(1,10\right))$ for $\beta_{RD}\;=\;15,R=6.642$ and D=0.001; (**g**) $p(S|S_{1}^{*}\;=\;\left(5,6\right))$ for $\beta_{RD}\;=\;0.1,R=0.001$ and D=4.965; (**h**) $p(S|S_{1}^{*}\;=\;\left(3,8\right))$ for $\beta_{RD}\;=\;0.65,R=0.789$ and D=3.287; (**i**) $p(S|S_{1}^{*}\;=\;\left(5,6\right))$ for $\beta_{RD}\;=\;15,R=6.642$ and D=0.001.

**Figure 2 entropy-22-01179-f002:**
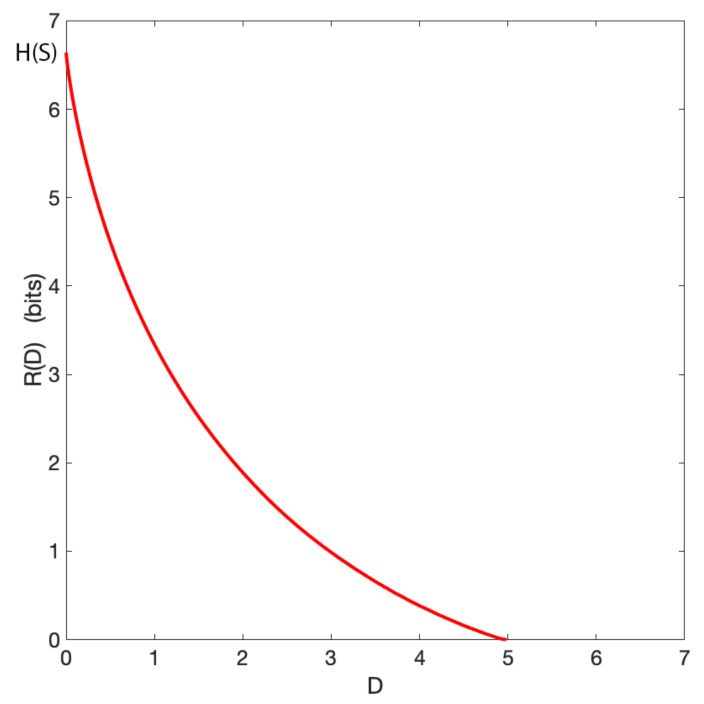
Geometric rate-distortion function of the grid world presented in Figure 1. The rate is denoted by R(D), with *D* indicating the corresponding distortion. H(S) denotes the entropy of the original space *S*.

**Figure 3 entropy-22-01179-f003:**
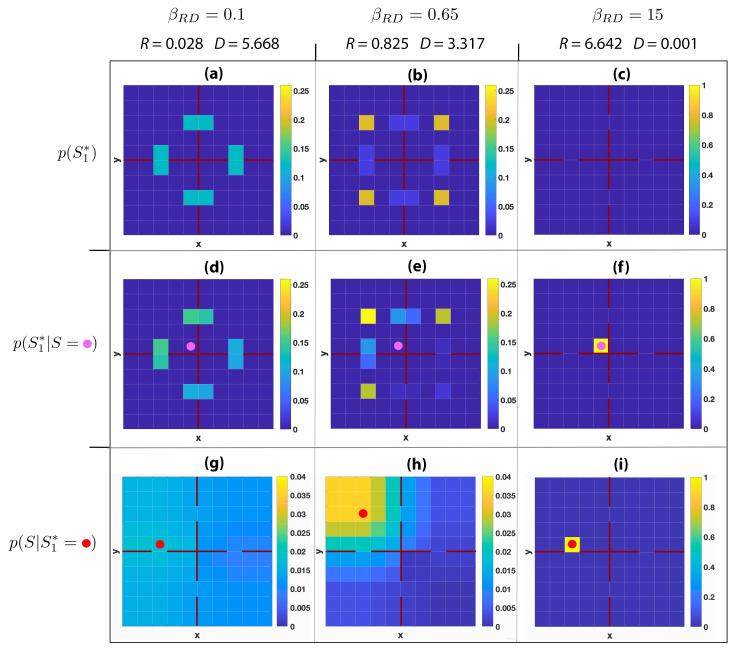
Geometric rate-distortion of a 10 × 10 grid world with a four-room topology. Each room has size 5 × 5, with room edges coloured in burgundy. Cells are coloured according to the probability of the reported distributions, with values indicated in the colour bars. Columns indicate the rates *R* and distortions *D* corresponding to different levels of trade-off, with $\beta_{RD}\;=\;0.1,0.65,15$. The first row reports values of the probability of picking a codeword $p(S_{1}^{*})$. The second row shows values of $p(S_{1}^{*}|S\;=\;s)$, that is the probability of selecting a codeword when the agent is in state *s*, where state s=(5,6) is represented by a magenta circle. The third row shows the values of $p(S|S_{1}^{*}\;=\;s_{1}^{*})$, which is the probability distribution of the agent’s position given that the codeword $s_{1}^{*}$ is chosen, where codeword locations $s_{1}^{*}$ are depicted by red circles. (**a**) $p(S_{1}^{*})$ for $\beta_{RD}\;=\;0.1,R=0.028$ and D=5.668; (**b**) $p(S_{1}^{*})$ for $\beta_{RD}\;=\;0.65,R=0.825$ and D=3.317; (**c**) $p(S_{1}^{*})$ for $\beta_{RD}\;=\;15,R=6.642$ and D=0.001; (**d**) $p(S_{1}^{*}|S\;=\;\left(5,6\right))$ for $\beta_{RD}\;=\;0.1,R=0.028$ and D=5.668; (**e**) $p(S_{1}^{*}|S\;=\;\left(5,6\right))$ for $\beta_{RD}\;=\;0.65,R=0.825$ and D=3.317; (**f**) $p(S_{1}^{*}|S\;=\;\left(5,6\right))$ for $\beta_{RD}\;=\;15,R=6.642$ and D=0.001; (**g**) $p(S|S_{1}^{*}\;=\;\left(3,6\right))$ for $\beta_{RD}\;=\;0.1,R=0.028$ and D=5.668; (**h**) $p(S|S_{1}^{*}\;=\;\left(3,8\right))$ for $\beta_{RD}\;=\;0.65,R=0.825$ and D=3.317; (**i**) $p(S|S_{1}^{*}\;=\;\left(3,6\right))$ for $\beta_{RD}\;=\;15,R=6.642$ and D=0.001.

**Figure 4 entropy-22-01179-f004:**
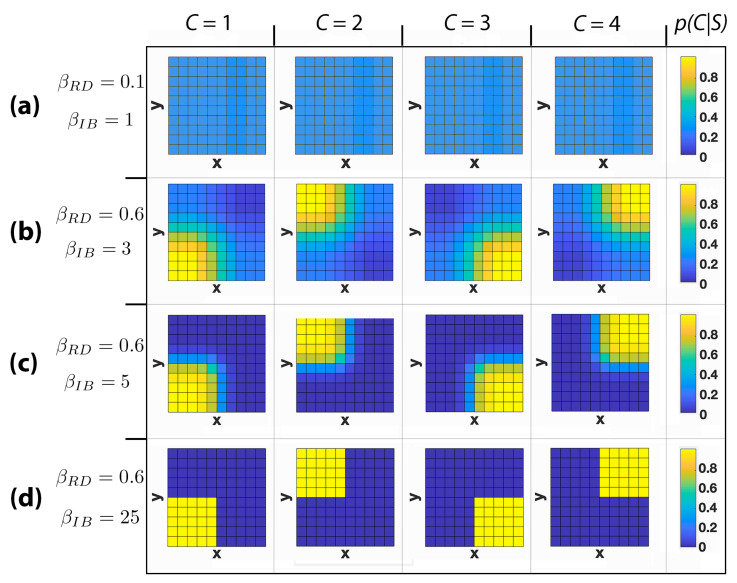
Clustering obtained when using the information bottleneck $C⇐S\to S_{1}^{*}$ for a 10 × 10 grid world. *S* denotes the original space, $S_{1}^{*}$ represents the compressed space, and *C* indicates the set of clusters. Cells are coloured according to the probability distribution p(C|S), with values reported in the colour bars. Each column represents a different cluster. Each row shows a different combination of the geometric rate-distortion and information bottleneck Lagrangian multipliers, denoted by $\beta_{RD}$ and $\beta_{IB}$ respectively. (**a**) $\beta_{RD}=0.1$ and $\beta_{IB}=1$; (**b**) $\beta_{RD}=0.6$ and $\beta_{IB}=3$; (**c**) $\beta_{RD}=0.6$ and $\beta_{IB}=5$; (**d**) $\beta_{RD}=0.6$ and $\beta_{IB}=25$.

**Figure 5 entropy-22-01179-f005:**
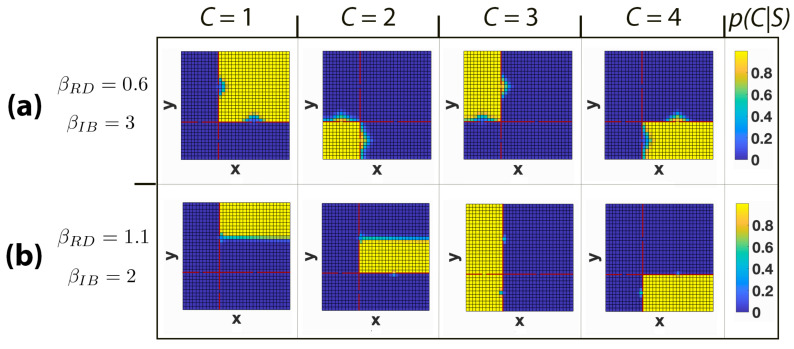
Clustering obtained when using the information bottleneck $C⇐S\to S_{1}^{*}$ for a 32 × 32 grid world with a four-room topology. *S* represents the original space, $S_{1}^{*}$ indicates the approximate space, and *C* denotes the set of clusters. Room edges are coloured in burgundy. States are coloured according to the probability values of p(C|S). Each column represents a different cluster. The two rows report different combinations of $\beta_{RD}$ and $\beta_{IB}$, denoting the geometric rate-distortion and information bottleneck trade-off parameters respectively. (**a**) $\beta_{RD}=0.6$ and $\beta_{IB}=3$; (**b**) $\beta_{RD}=1.1$ and $\beta_{IB}=2$.

**Figure 6 entropy-22-01179-f006:**
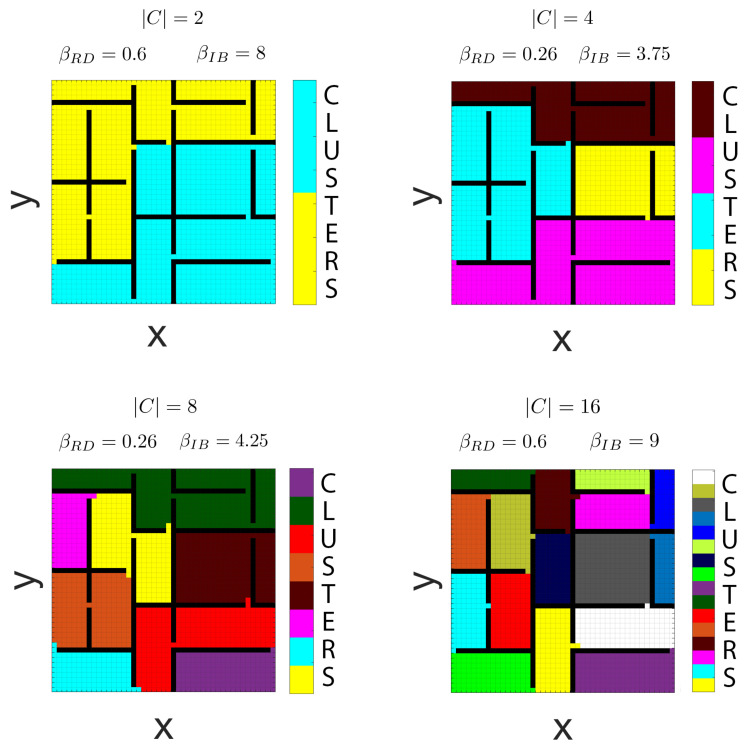
Hard clustering obtained using the information bottleneck $C⇐S\to S_{1}^{*}$ for a 45 × 45 grid world with a 16-room topology. *S* denotes the original space, $S_{1}^{*}$ represents the compressed space, and *C* indicates the set of clusters. Each grid represents the partitioning obtained using a different number of clusters, with |C|=2,4,8,16 respectively. Cells are coloured according to the identity of the cluster they belong to and the clusters’ colours are depicted in the colour bars. $\beta_{RD}$ and $\beta_{IB}$ indicate the geometric rate-distortion and information bottleneck Lagrangian multipliers respectively.

**Figure 7 entropy-22-01179-f007:**
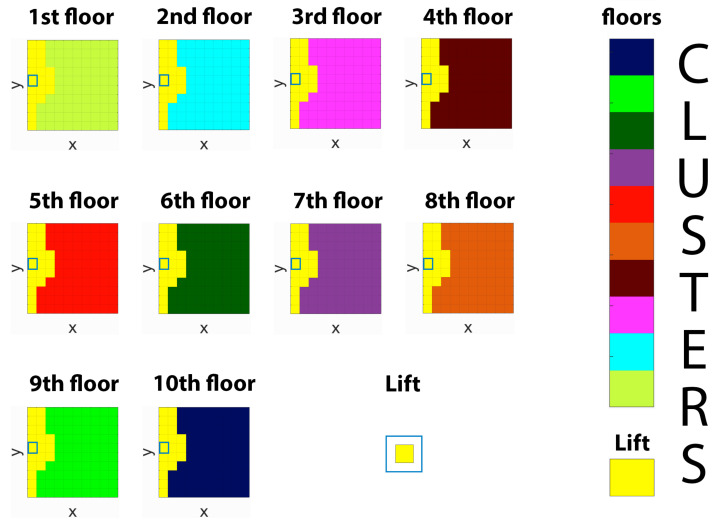
Hard clustering obtained with the information bottleneck $C⇐S\to S_{1}^{*}$ in a multi-floor scenario with |C|=11. *S* represents the original space, $S_{1}^{*}$ indicates the approximate space and *C* denotes the set of clusters. Floors are 10 × 10 grids connected to each other with a lift. The lift’s entrance of each floor is located at cell (1,5), as indicated by blue-framed squares. The lift is represented by a separate cell. Cells are coloured according to the identity of the cluster they belong. Clusters’ colours are shown in the colour bars.

**Figure 8 entropy-22-01179-f008:**
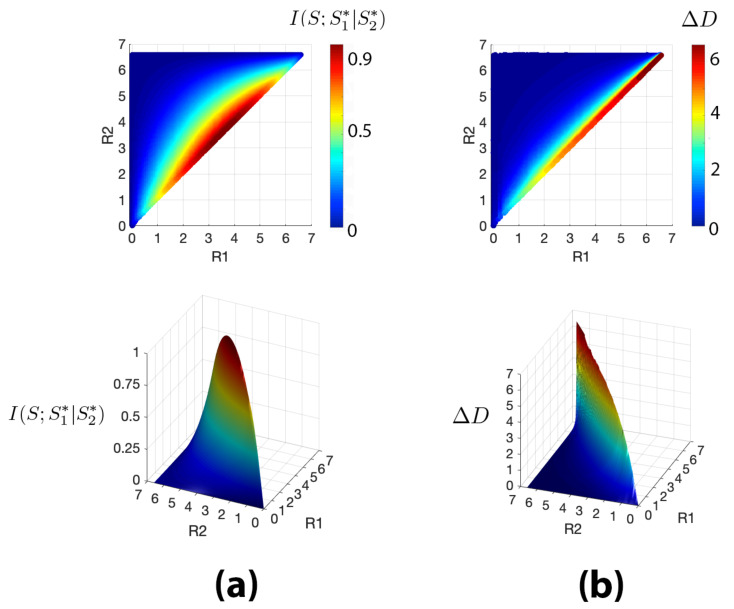
(**a**) Colour and surface plots of the mutual information $I(S;S_{1}^{*}|S_{2}^{*})$ embedded in the $\left(R_{1},R_{2}\right)$ plane. *S* denotes the original space, $S_{1}^{*}$ represents the first level of refinement constructed at rate $R_{1}$, and $S_{2}^{*}$ indicates the second level of refinement obtained with rate $R_{2}$. The magnitude of $I(S;S_{1}^{*}|S_{2}^{*})$ is used as measure of soft Markovianity; (**b**) Colour and surface plots of the successive refinement distortion error $\Delta D\;=\;{\tilde{D}}_{2}-D_{2}$ embedded in the $\left(R_{1},R_{2}\right)$. ${\tilde{D}}_{2}$ denotes the total distortion incurred with successive refinement and $D_{2}$ is the distortion incurred when geometric rate-distortion is done directly at rate $R_{2}$. All plots are reported for rates such that $R_{2}>R_{1}$.

**Figure 9 entropy-22-01179-f009:**
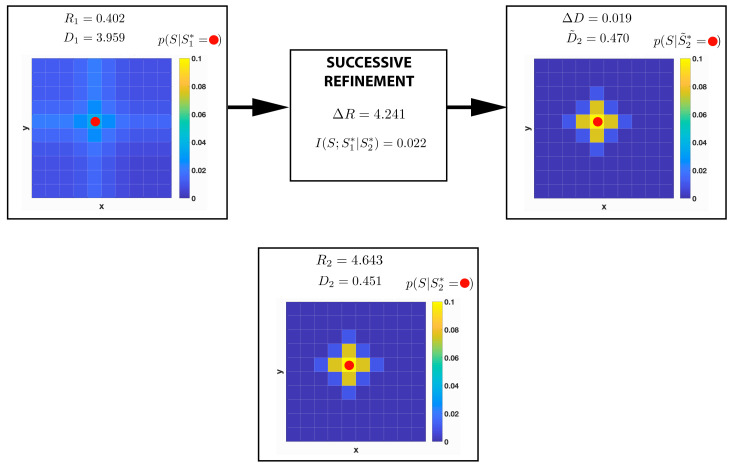
Schematic representation of one instance of successive refinement for a 10 × 10 grid world. Different levels of refinement are represented by grid worlds with cells coloured according to $p(S|S_{i}^{*}=s)$, with i=1,2 and s=(5,6) (shown as a red circle). *S* represents the original space, $S_{1}^{*}$ indicates the first level of refinement, and $S_{2}^{*}$ denotes the second level of refinement. The first approximate spatial representation, which is depicted on the left, is obtained using Geometric Rate-Distortion (GRD) at rate $R_{1}=0.402$ bits with distortion $D_{1}=3.959$. The successive representation is obtained refining the first one at rate $\Delta R=R_{2}-R_{1}=0.022$ bits and is shown on the right. The bottom grid shows the spatial representation obtained via GRD at rate $R_{2}=R_{1}+\Delta R=4.643$ bits with distortion $D_{2}=0.451$. $\Delta D={\tilde{D}}_{2}-D_{2}$, where ${\tilde{D}}_{2}$ denotes the total distortion incurred using successive refinement. The mutual information $I(S;S_{1}^{*}|S_{2}^{*})$ is used as measure of soft Markovianity.

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
