# Peer review of "Space Emerges from What We Know—Spatial Categorisations Induced by Information Constraints"

_entropy, 2020, doi:10.3390/e22101179_

Round 1

Reviewer 1 Report

This work focus on space representation by biologically plausible agents. The authors combine a "geometric" rate-distortion function with the information bottleneck method to model the spatial representation of the agent’s environment. By using the successive refinement method they exemplify how the spatial representations can be incrementally refined without losing previous acquired information.

The assumptions of the work are fairly funded on biological principles and the manuscript is well written and easy to follow. I believe this work is an original contribution that fits well with the aims of the journal and that I consider worth of publication. I only have few suggestions and additional minor comments below to improve the manuscript.

The Abstract requires some additions/modifications. The topic and the methods are well introduced but there is no mention to what the main finding is and what it means for the specific field.

At the end of introduction the authors state that they will end the manuscript including a perspectives on future work that I only see alluded to in the Conclusions. I think few more words on the importance of their work and how concretely they suggest to continue this line of investigation will help strengthen the message.

To make the figures self-explanatory all expressions should be defined. For example in Fig. 1 it should be written that p(S∗1) is the Geometric Rate-Distortion output distribution and so on for the other expressions. Furthermore, R and D should also be defined in the legend.

Minor comments

Line 171. The "i.i.d." is not defined before and might not be a familiar abbreviation for everyone.

Figure 1 Legend. After 0.65 there should be a coma and not a dot.

In the legend of Fig. 3 rooms' edges are magenta but in the text (line 368) are burgundy. Please change the first or the latter.

Author Response

We thank you for the useful comments. We agreed with your suggestions.

The text has been modified taking into account what you suggested, including what you indicated in the “minor comments”. In what follows we have briefly indicated how we have tackled the raised points. To have more details, please refer to the modified text, where text coloured in red represents what has been modified or added.

Reviewer 1: “The Abstract requires some additions/modifications. The topic and the methods are well introduced but there is no mention to what the main finding is and what it means for the specific field.”

Authors: Keeping in mind that an Abstract cannot be too long, we have extended the Abstract including a summary of the main findings of the paper and why we believe these are significant.

Reviewer 1: “At the end of introduction the authors state that they will end the manuscript including a perspectives on future work that I only see alluded to in the Conclusions. I think few more words on the importance of their work and how concretely they suggest to continue this line of investigation will help strengthen the message.”

Authors: We have added new paragraphs to the Conclusion section that indicates additional possible future developments of the presented work. We think that these are relevant and underline the importance of our study.

Reviewer 1: “To make the figures self-explanatory all expressions should be defined. For example in Fig. 1 it should be written that p(S1) is the Geometric Rate-Distortion output distribution and so on for the other expressions. Furthermore, R and D should also be defined in the legend.”

Authors: We have added in every figure’s legend a brief definition of all the variables and expressions reported in the figures.

Reviewer 2 Report

This is a rather interesting study, yet there is an imbalance between the very ambitious introduction and the reminder of the paper: The introduction  surveys cases of experiential spaces whose structure is a function of their use, ranging from birds’ navigation, through London’s tube map, to Polynesian sea navigators. This creates an expectation that next comes a conception of space that captures all these. Yet this is not the case. The mathematics in the body of the text suggests something much more modest and limited – to my mind, since there are aspect of subjective experiential space that do not lent themselves to mathematical quantification. Weather this is indeed so, or a consequence of the current technological limitation, remain to be seen. In any event the issue should be addressed in the paper.

Author Response

Thank you indeed for your comments, we actually quite agree with them
- indeed, we realized that we opened a major set of questions with our
introduction and when our current technical developments only denote
inroads towards where we are trying to go. We also considered whether
it might have been prudent to cut out the large-scale picture
entirely, however, we also felt that we wanted to give the reader a
larger-scale idea where we are trying to travel towards. We felt that
this would be important to judge context of the work.

Now, of course, as long as a comprehensive proof is still outstanding,
it will be difficult to argue that the formalism is indeed capable of
modeling the whole set of listed phenomena. Nonetheless, we feel there
are already now some promising indications that informational
formalisms are indeed able to capture more than one would naively
expect and move towards the envisaged concepts.

For instance, we had mentioned in the intro the goal-relevant
information/subgoals, briefly mentioned in the introduction, of course
the spatial structuring; some of the geodesic representation mentioned
in the literature would immediately into our formalism, even if we
haven't done it explicitly, and - most dominantly - our paper's
contribution in terms of the the successive refinement which is an -
almost purely informational - concept and, to our knowledge, never
before used outside of information theory and image compression. The
work by Haun et al. 2019, also mentioned in the intro, shows that
there are already information theory-inspired inroads into modelling
the coexistence of different spatial concepts. Thus, we believe that
this paper brings to an already existing table an additional number of
tools and indications that it is a worthwhile enterprise to try to
drive this line of questions further.

However, we quite agree with the overly raised expectations in the
introduction. Therefore, to put things in perspective more clearly, we
have therefore added a section to the beginning of the introduction
and a paragraph to the end to make clear that the reader should not
expect a final picture at this stage.

We hope that the new formulation now achieves an acceptable balance
between mapping the agenda and the concrete technical tools brought to
the table at present.